# Intricate Relations Among Particle Collision, Relative Motion and Clustering in Turbulent Clouds: Computational Observation and Theory.

Ewe-Wei Saw [1,2] and Xiaohui Meng [1]

[1]School of Atmospheric Sciences and Guangdong Province Key Laboratory for Climate Change and Natural Disaster Studies, Sun Yat-Sen University, Zhuhai, China
[2]Ministry of Education Key Laboratory of Tropical Atmosphere-Ocean System, Zhuhai, China

**Correspondence:** E.-W. Saw (ewsaw3@gmail.com), X. Meng (mengxh7@mail2.sysu.edu.cn)

**Abstract.**

Considering turbulent clouds containing small inertial particles, we investigate the effect of particle collision, in particular collision-&-coagulation, on particle clustering and particle relative motion. We perform direct numerical simulation (DNS) of coagulating particles in isotropic turbulent flow in the regime of small Stokes number ($St = 0.001 - 0.54$) and find that, due to collision-coagulation, the radial distribution functions (RDF) fall-off dramatically at scales $r \sim d$ (where $d$ is the particle diameter) to small but finite values; while the mean radial-component of particle relative velocities (MRV) increase sharply in magnitudes. Based on a previously proposed Fokker-Planck (drift-diffusion) framework, we derive a theoretical account of the relationship among particle collision-coagulation rate, RDF and MRV. The theory includes contributions from turbulent-fluctuations absent in earlier mean-field theories. We show numerically that the theory accurately accounts for the DNS results (i.e. given an accurate RDF, the theory could produce the accurate MRV). Separately, we also propose a phenomenological model that could directly predict MRV and find that it is accurate when calibrated using 4th moments of the fluid velocities. We use the model to derive a general solution of RDF. We uncover a paradox: the past empirical success of the differential version of the theory is theoretically unjustified. We see a further shape-preserving reduction of the RDF (and MRV) when the gravitational settling parameter ($S_g$) is of order $O(1)$. Our results demonstrate strong coupling between RDF and MRV and imply that earlier isolated studies on either RDF or MRV have limited relevance for predicting particle collision rate.

## 1   Introduction

The motion and interactions of small particles in turbulence have fundamental implications for atmospheric clouds, specifically, it is relevant to the time-scale of rain formation particularly in warm-clouds (Falkovich et al., 2002; Wilkinson et al., 2006; Grabowski and Wang, 2013) [a similar problem also applies to planet formation in astrophysics (Johansen et al., 2007)]. It is also important for engineers who are designing future, greener, combustion engines, as this is a scenario they wish to understand and control in order to increase fuel-efficiency (Karnik and Shrimpton, 2012). Cloud particles or droplets, due to their inertia, are known to be ejected from turbulent vortices and thus form clusters – regions of enhanced particle-density (Wood et al.,

2005; Bec et al., 2007; Saw et al., 2008; Karpińska et al., 2019); this together with droplet collision is of direct relevance for the mentioned applications. Due to the technical difficulty of obtaining extensive and systematic experimental or field data on particle/droplet collision in turbulent cloud, many of the recent studies rely on direct numerical simulation (DNS), examples of which could be found in e.g. (Onishi and Seifert, 2016; Wang et al., 2008) and reference therein. Up until now, we do not have definitive answers to basic questions such as how to calculate particle collision rate from basic turbulence-particle parameters and what is the exact relation between collision and particle clustering and/or motions, for, as we shall see, our work reveals that collision-coagulation causes profound changes in particle relative velocity statistics and particle clustering, questioning earlier understanding of the problem. The difficulty of this problem is in part related to the fact that turbulence is, even by itself, virtually intractable theoretically due to its nonlinear and complex nature.

The quest for a theory of particle collision in turbulence started in 1956 when Saffman and Turner (1956) derived a mean-field formula for collision rate of finite size, inertialess, particles. In another landmark work (Sundaram and Collins, 1997), a general relation among collision-rate ($R_c$), particle clustering and mean particle relative radial velocity was presented: $R_c/(n_1 n_2 V) = 4\pi d^2 g(d) \langle w_r(d) \rangle_*$, where $g(r)$ is the particle radial distribution function (RDF), $w_r$ is the radial component of relative velocity between two particles, $\langle \cdot \rangle_*$ denotes averaging over particle-pairs, $\langle w_r(d) \rangle_*$ is the mean radial-component of relative particle velocity (MRV), $n_i$'s are global averages of particle number density, $V$ is the spatial volume of the domain, $d$ the particle diameter. The remarkable simplicity of this finding inspired a "separation paradigm", which is the idea that one could study the RDF or MRV separately (which are technically easier), the independent results from the two may be combined to accurately predicts $R_c$ (an idea that we subsequently challenge). Another work of special interest here is the drift-diffusion model by Chun et al. (2005) (hereafter: CK theory) (note: there are other similar theories (Balkovsky et al., 2001; Zaichik and Alipchenkov, 2003)). The CK theory, derived for non-colliding particles in the limit of vanishing particle Stokes number $St$ (a quantity that reflects the importance of the particle's inertia in dictating its motion in turbulence), correctly predicted the power-law form of the RDF (Reade and Collins, 2000; Saw et al., 2008) and have seen remarkable successes over the years including the accurate account of the modified RDF of particles interacting electrically (Lu et al., 2010) and hydrodynamically (Yavuz et al., 2018).

Here, we first present results on RDF and MRV for particles undergoing collision-coagulation[1]. The data is obtained via direct numerical simulation (DNS), which is the gold-standard computational method in terms of accuracy and completeness for solving the most challenging fluid dynamics problem i.e. turbulent flows. It is worth noting that the focus of our work is on the fundamental relationship between collision, RDF and MRV, and to highlight differences from the case with non-colliding particles (Chun et al., 2005). To that end, we have designed the DNS to have an idealized setup similar to what was done in (Chun et al., 2005), which would allow us to identify without doubt the effects of particle collision-coagulation. As a result, this limits the direct applicability to real systems (these limitations are detailed in Sec. 4.5).

Analysis of the DNS results is followed by a theoretical account of the relations between collision-rate, RDF and MRV that includes mean-field contributions (Saffman and Turner, 1956; Sundaram and Collins, 1997) and contributions from turbulent

---

[1]Coagulation is, in a sense, the simplest outcome of collision. In the sequel we shall argue that the major qualitative conclusions of our work also apply to cases with other collisional outcomes.

fluctuations (absent from earlier theories (Saffman and Turner, 1956; Sundaram and Collins, 1997)). The theory is derived from the Fokker-Planck (drift-diffusion) framework first introduced in the CK theory (Chun et al., 2005). We shall see that the main effect of collision-coagulation is the enhanced asymmetry in the particle relative velocity distribution[2] and that this leads to nontrivial outcomes.

## 2   Direct Numerical Simulation (DNS)

To observe how particle collision-coagulation affects RDF and MRV, we performed direct numerical simulation (DNS) of steady-state isotropic turbulence embedded with particles of finite but sub-Kolmogorov size. We solve the incompressible Navier-Stokes Equations (Eq. (1)) using the standard pseudo-spectral method (Rogallo, 1981; Pope, 2000; Mortensen and Langtangen, 2016) inside a triply periodic cubic-box.

$$\frac{\partial \mathbf{u}}{\partial t} + \mathbf{u} \cdot \nabla \mathbf{u} = -\frac{1}{\rho}\nabla p + \nu \nabla^2 \mathbf{u} + \mathbf{f}(\mathbf{x}, t),$$

$$\nabla \cdot \mathbf{u} = 0, \tag{1}$$

where $\rho, p, \nu, \mathbf{f}$ are the fluid mass-density, pressure, kinematic viscosity, imposed forcing respectively. The velocity field is discretized on a $256^3$ grid. Aliasing resulting from Fourier transform of truncated series is removed via a 2/3-dealiasing rule (Rogallo, 1981). A statistically stationary and isotropic turbulent flow is achieved by continuously applying random forcing to the lowest wave-numbers until the flow's energy spectrum is in steady-state (Eswaran and Pope, 1988). The 2nd-order Runge-Kutta time stepping was employed. Further details of such a standard turbulence simulator can be found in e.g. (Pope, 2000; Rogallo, 1981; Mortensen and Langtangen, 2016). The accuracy of DNS for turbulent flows have been experimentally validated for decades (see e.g. the compilation of results in Pope (2000));

Particles in the simulations are advected via a viscous Stokes drag force (Maxey and Riley, 1983):

$$d\mathbf{v}/dt = (\mathbf{u} - \mathbf{v})/\tau_p,$$

where $\mathbf{u}, \mathbf{v}$ are the local fluid and particle velocity respectively, $\tau_p$ is the particle inertia response time, defined as $\tau_p = \frac{1}{18}(\rho_p/\rho - 1)(d^2/\nu)$, where $\rho_p$ is the particle mass-density and $d$ is the particle diameter. As mentioned, this work focuses on the fundamental relationship between collision-coagulation, RDF and MRV, as well as on addressing the validity of the theory (to be described). It is thus, beneficial to keep the DNS setting idealized (and in the regime relevant for the theory) for the sake of clarity when interpreting results. To that end, the DNS does not include inter-particle hydrodynamic interactions (HDI) and gravitational settling, nor does it consider the effects of temperature-, humidity-variation and phase transitions. Such practice is not uncommon in studies designed to isolate and address fundamental issues related to particles dynamics in turbulence, examples that are closely related to the current setup and/or problem include (Sundaram and Collins, 1997; Chun et al., 2005; Bec et al., 2007; Salazar et al., 2008; Wang et al., 2008; Woittiez et al., 2009; Voßkuhle et al., 2013). However, such an

---

[2]In the collision less case, the asymmetry is much weaker and is related to viscous dissipation of energy in turbulence (Pope, 2000).

| $Re_\lambda$ | $\nu$ [dm$^2$/s] | $u_{rms}$ [dm/s] | $\epsilon$ [dm$^2$/s$^3$] | $\eta$ [dm] | $\tau_\eta$ [s] | $L_c$ [dm] | $d$ [dm] |
|---|---|---|---|---|---|---|---|
| 133 | 0.001 | 0.613 | 0.117 | 0.00962 | 0.0925 | $2\pi$ | $d_*$ or $2d_*$ |

**Table 1.** Values of the parameters in the DNS. (Note: dm = decimeter). From the left, we have the Taylor-scale Reynolds number, kinematic viscosity of the fluid, root-mean-square of fluid velocity, kinetic energy dissipation rate, Kolmogorov length- and time-scale, length of the simulation cube and particle diameters considered. We have introduced $d_*$ to represent the specific value: $9.49 \times 10^{-4}$dm (more details in the text). We choose the units of the length (time) scale in the DNS to be in decimeter (second), such that $\nu$ is nearly its typical value in the atmosphere.

approach certainly limits the direct applicability of our results to some realistic problems in the atmosphere, these limitations will be detailed in Sec. 4.5, where a discussion of the effects of gravity and HDI is also given.

In this context, the particle Stokes number, defined as $\tau_p/\tau_\eta$ where $\tau_\eta$ is the Kolmogorov time-scale, could be expressed as $St = \frac{1}{18}(\rho_p/\rho - 1)(d/\eta)^2$, where $\eta$ is the Kolmogorov length-scale. Time-stepping of the particle motion is done using a 2nd-order modified Runge-Kutta method with "exponential integrator" that is accurate even for $\tau_p$ much smaller than the fluid's time-step (Ireland et al., 2013). The particles introduced into the simulation are spherical and are of the same size, the initial number of particles is $10^7$ and they are randomly distributed in space. Particles collide when their volumes overlap and a new particle is formed conserving volume and momentum (Bec et al., 2016). We continuously, randomly, inject new particles so that the system is in a steady-state after some time. Statistical analysis is done at steady-state on monodisperse particles (i.e. particles with the same $St$). Experimental validation of the accuracy of such particle simulating scheme in DNS could be found in Salazar et al. (2008); Saw et al. (2012, 2014); Dou et al. (2018).

Values of key parameters of the DNS are given in Table 1. Values of other parameters and further details could be found in (Supplements).

## 3 Elements of the Drift-Diffusion Theory

As described in (Chun et al., 2005), in the limit of $St \ll 1$, particle motions are closely tied to the fluid velocity and, to leading order, completely specified by the particle position and fluid velocity gradients. We consider the Fokker-Planck equation which is closed and deterministic (see e.g. Appendix J in (Pope, 2000)):

$$\frac{\partial P}{\partial t} + \frac{\partial (W_i P)}{\partial r_i} = 0, \tag{2}$$

where $P \equiv P(r_i, t \,|\, \Gamma_{ij}(t))$ is the (per volume) probability density (PDF) for a secondary particle to be at vector position $r_i$ relative to a primary particle at time $t$, conditioned on a fixed and known history of the velocity gradient tensor along the primary particle's trajectory $\Gamma_{ij}(t)$, $W_i$ is the mean velocity of secondary particles relative to the primary, under the same condition. Note: $W_i$ is a conditional-average, while $w_i$ denotes a realization of relative velocity between two particle.

From this, one could derive an equation for $\langle P \rangle$:

$$\frac{\partial \langle P \rangle}{\partial t} + \frac{\partial}{\partial r_i} \left( \langle W_i \rangle \langle P \rangle + \langle W_i P' \rangle \right) = 0, \tag{3}$$

where $\langle . \rangle$ implies ensemble averaging over primary particle histories (note: $\langle W_r \rangle \equiv$ unconditional mean of $w_r$, averaged over all particle pairs, i.e. the MRV). This equation, however, is not closed due to the correlation between the fluctuating terms $W_i$ and $P' \equiv P - \langle P \rangle$. The correlation $\langle W_i P' \rangle$ can be written in terms of a drift flux and diffusive flux (detailed derivation is well described in (Chun et al., 2005)), such that we have:

$$\frac{\partial \langle P \rangle}{\partial t} + \frac{\partial}{\partial r_i} \left( q_i^d + q_i^D \right) + \frac{\partial (\langle W_i \rangle \langle P \rangle)}{\partial r_i} = 0, \tag{4}$$

where the drift flux is:

$$q_i^d = -\int_{-\infty}^{t} \left\langle W_i(\mathbf{r}, t) \frac{\partial W_l}{\partial r_l'}(\mathbf{r}', t') \right\rangle \langle P \rangle (\mathbf{r}', t') \, dt', \tag{5}$$

and the diffusive flux is:

$$q_i^D = -\int_{-\infty}^{t} \langle W_i(\mathbf{r}, t) W_j(\mathbf{r}', t') \rangle \frac{\partial \langle P \rangle}{\partial r_j'}(\mathbf{r}', t') \, dt', \tag{6}$$

where $\mathbf{r}'$ satisfies a characteristic equation: $\frac{\partial r_i'}{\partial t'} = W_i(\mathbf{r}', t')$, with boundary condition: when $t' = t$, $r_i' = r_i$.

Finally we note that, since particles are allowed to collide-coagulate in our theory, we use the conventional definition of MRV: $\langle W_r \rangle \equiv \langle w_r \rangle_*$. In some works that consider non-colliding (ghost) particles, the conditional mean $\langle w_r \mid w_r \leq 0 \rangle_*$ must be used for the purpose of calculating mean collision rate, since $\langle w_r \rangle_* = 0$ in isotropic turbulence (Chun et al., 2005).

## 4 DNS Results, Theory and Discussion

We compute the RDF via $g(r) = N_{pp}(r) / [\frac{1}{2} N(N-1)\delta V_r / V]$, where $N_{pp}(r)$ is the number of particle pairs found to be separated by distance $r$, $\delta V_r$ is the volume of a spherical shell of radius $r$ and infinitesimal thickness $\delta r$,

Figure 1 shows the RDFs obtained for monodisperse particles of various Stokes numbers and sizes. Two cases ($St = 0.22$ and 0.54) are shown in panel-a and two more ($St = 0.054$ and 0.001) are shown in panel-b. In this work, we focus on the smaller values of $St$ since the theory which we shall consider is also only applicable in the $St \ll 1$ regime. However, we have included the $St = 0.54$ case to demonstrate that the observations to be described extends also to finite $St$. In all cases, except one, the particles are of the same size $d = d_*$, where $d_*$ represents the specific value of $d_* = 9.49 \times 10^{-4}$ dm, chosen so that the particle sizes are about $O(0.1)$ times the Kolmogorov scale ($\eta$), thus allowing us to still observe a regime ($3d \lesssim r \lesssim 30\eta$) of power-law RDFs. To shows the effect of changing particle size, panel-a also includes a case of ($St = 0.54, d = 2d_*$) for comparison. Looking at panel-a, apart from the apparent power-law behavior of the RDFs at intermediate values of $r$, the most striking feature of these RDFs for colliding-coagulating particles is that they fall-off dramatically in the $r \sim d$ regime. This is

very different from what was seen in earlier studies of non-colliding particles where $g(r)$ are simple power-laws (Chun et al., 2005; Saw et al., 2008). We also see that as $r$ approaches $d$ the steepness of the curve (see e.g. the blue-circles) increases as $g(r)$ drops-off, this and the fact that the abscissa is logarithmic implies that $\frac{\partial g}{\partial r}$ is increasing exponentially in the process. As a consequence, it is difficult to discern from these plots if the limit of $g(r)$ at particle contact ($r \to d$) is still nonzero. This is an important question as $\lim_{r \to d}[g(r)]=0$ implies that the mean-field formula of Sundaram and Collins (1997) has zero contribution towards $R_c$, i.e. collision rate is solely due to turbulent-fluctuations. It is only by re-plotting $g(r)$ versus $r-d$ (see insets in Fig. 1), and using a remarkable resolution that is $10^3$ finer than $d$, that we see a convincing trend supporting a finite $g(r \to d)$. Also clear in panel-a is the observation that with changing particle size ($d$) the location of the sharp fall-off merely shifts to where the new value of $d$ is.

The strong effect of particle collision on the RDF (also on MRV as we shall see later) challenges the validity of the "separation paradigm". We note that similar fall-off of RDF was previously observed (Sundaram and Collins, 1997) but a complete analysis and theoretical understanding were lacking. Also, a study on multiple collisions (Voßkuhle et al., 2013) had hinted at the potential problem with the separation paradigm.

Another observation is that in the power-law regime ($3d \lesssim r \lesssim 30\eta$), the RDFs appear (as expected) as straight-lines with slopes (i.e. power-law exponents) that increase with $St$ and are numerically consistent with those found for non-colliding particles (see e.g. (Saw et al., 2012)).

### 4.1 Theoretical Account via Drift-Diffusion Theory

To theoretically account for the new findings, we make some derivations that are partially similar to the ones in (Chun et al., 2005), but under a new constraint due to coagulations: At contact ($r = d$), the radial component of the particle relative velocities can not be positive[3], while with increasing $r$ the constraint is gradually relaxed. The first consequence of this is that the distribution of the radial component of the relative particle velocity ($W_r$) is highly asymmetric at $r \approx d$, i.e. the PDF of positive $W_r$'s are very small (this constitutes the "enhanced asymmetry" mentioned earlier). Thus for $r \approx d$, $\langle W_r \rangle$ must be negative. In Sec. 3, we showed that in the $St \ll 1$ limit, one could derive a master equation (Eq. 4), reproduced here for clarity:

$$\frac{\partial \langle P \rangle}{\partial t} + \frac{\partial}{\partial r_i} \left( q_i^d + q_i^D \right) + \frac{\partial (\langle W_i \rangle \langle P \rangle)}{\partial r_i} = 0 \, ,$$

where $q_i^d$ is the drift flux (of probability due to turbulent fluctuation) defined in (5) and $q_i^D$ is the diffusive flux defined in (6).

We then expand $W_i$, $\frac{\partial W_l}{\partial r_l}$ and (consequentially) the fluxes as perturbation series with $St$ as the small parameter (details in (Supplements) or (Chun et al., 2005)). The coagulation constraint affects the values of the coefficients of these series. For the drift flux, the leading order terms (in powers of $St$) are:

$$q_i^d = - \langle P \rangle (\mathbf{r}) \, r_k \int_{-\infty}^{t} \left[ a_{ik}^{(1)} St + a_{ki}^{(2)} St^2 \right] dt' \, , \tag{7}$$

---

[3] In other words particles may approach each other (and collide) but they can not be created at contact and then separate.

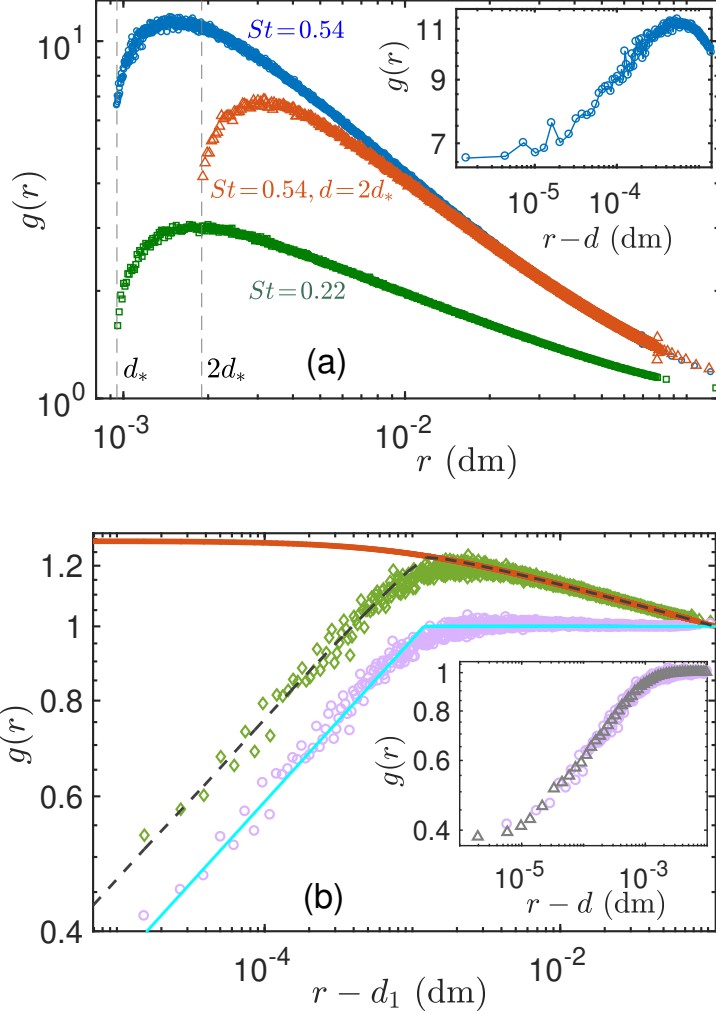

**Figure 1.** RDFs ($g(r)$) of particles that coagulate upon collision. **a)** $g(r)$ for various cases of Stokes numbers and particle diameters ($d$). $\square$: $St=0.22$, $d=d_*$, $\bigcirc$: $St=0.54$, $d=d_*$, $\triangle$: $St=0.54$, $d=2d_*$. All $g(r)$ drop-off exponentially when $r \to d$ (details in text). **Inset:** $g(r)$ versus $r-d$ for the $\bigcirc$ case. It exemplifies the fact that $\lim_{r\to d} g(r)$ is nonzero. **b)** RDFs versus $r-d_1$ (where $d_1=0.99d$) for the case of $St=0.054$, $d=d_*$. $\diamond$: the raw DNS-produced RDF ($g_{\mathrm{DNS}}(r)$). Red-line: power-law fit to $g_{\mathrm{DNS}}(r)$ (i.e. the $\diamond$-plot) in the large-$r$ regime (the fit result is $0.890r^{-0.0535}$). It is equivalent to $g_s(r)$ in the ansatz $g_a(r) = g_0(r)g_s(r)$, i.e. it is the expected RDF for non-colliding particles under the same conditions. $\bigcirc$: the compensated RDF, i.e. $g_{\mathrm{DNS}}(r)/g_s(r)$ (note: $g_s(r)$ is the red-line described earlier), this essentially gives us $g_0(r)$, which may be understood as a 'modulation' on the RDF due to collision-coagulation. Cyan-line: two-piece power-law fits to the compensated RDF (the $\bigcirc$-plot) in the small and large $r-d_1$ regimes respectively (fit results: $4.17(r-d_1)^{0.212}$, $1.00(r-d_1)^{-2\times10^{-4}}$), this is an estimate for $g_0(r)$. Black-dashed-line: $g_0(r)g_s(r)$, (cyan-line × red-line), this shows that the ansatz accurately reproduces $g_{\mathrm{DNS}}(r)$. **Inset:** RDFs versus $r-d$. $\bigcirc$: compensated $g(r)$ for $St=0.054$, $d=d_*$, equivalent to the $\bigcirc$-plot in the panel's main figure; $\triangle$: compensated RDF for case $St=0.001$, $d=d_*$, i.e. finite size, almost zero $St$ particles (in this case, the compensated and raw RDFs are the identical). This inset suggests that $g_0(r)$ has negligible St-dependence.

with $a_{ik}^{(1)} = \tau_\eta \langle \Gamma_{ik}(t)\Gamma_{lm}(t')\Gamma_{ml}(t')\rangle$ and $a_{ki}^{(2)} = \tau_\eta^2 \langle \Gamma_{ij}(t)\Gamma_{jk}(t)\Gamma_{lm}(t')\Gamma_{ml}(t')\rangle$, $\Gamma_{ij}$ is the $ij$-th component of the fluid's velocity gradient tensor at the particle position (the $a_{ik}$'s are thus related to two-time correlations of moments of velocity gradients, Chun et al. (2005) shown that $a_{ik}^{(2)} \propto \overline{S^2} - \overline{R^2}$, where $(\overline{S^2}, \overline{R^2})$ are the average fluid (strain rate tensor, rotation rate tensor) squared at particle positions). As explained earlier, coagulation-constraint causes the PDF of relative particle velocities to become highly asymmetric for $r \sim d$, thus $a_{ik}^{(1)}$ is nonzero at this scale. This is very different from the case of non-colliding particles (Chun et al., 2005) where $a_{ik}^{(1)}$ is always zero due to statistical isotropy. Under the constraint, DNS gives $\int_{-\infty}^{t} a_{ik}^{(1)} dt' \approx -0.18\,s^{-1}$ and $\int_{-\infty}^{t} a_{ki}^{(2)} dt' \approx 2.45\,s^{-1}$ (more in (Supplements)). Thus for $r \sim d$, the drift flux is negative for large $St$ but becomes positive[4] when $St$ is below the value of $\approx 0.07$; and in the limit of $St \to 0$, it is dominated by the first term in (7).

$q_i^D$ is a 'nonlocal' diffusion caused by fluctuations and can be estimated using a model that assumes the particle relative motions are due to a series of random uniaxial straining flows (Chun et al., 2005). Chun et al. (2005) showed that, generally, $q_i^D$ has an integral form (due to nonlocality), and only in the special case where $g(r)$ is a power-law, may it be cast into a differential form (similar to a local diffusion). In view of the nontrivial $g(r)$ observed here, we must proceed with the integral form:

$$q_r^D = c_{st}\, r \int d\Omega \int_0^\infty dt_f F(t_f) \int_{d/r}^\infty dR_0\, R_0{}^2 \langle P \rangle (r R_0)\, f_I(R_0, \mu, t_f),$$

where $R_0 \equiv r_0/r$ with $r_0$ as the initial separation distance of a particle pair before a straining event; $F$ the probability density function for the duration of each event; $f_I$ is determined by relative prevalence of extensional versus compressional strain events (more details in (Supplements) or (Chun et al., 2005)); $\Omega$ is the solid angle for the axis of the straining flow; note: due to coagulation, the $R_0$-integration starts from $d/r$. We differ crucially from the CK theory via the introduction of the (positive) factor $c_{st}$, which could be shown to equal $|c_1|$, where $c_1$ is the power law exponent of the RDF the particles would have assuming they are non-colliding (details in (Supplements)).

By definition, $g(r) \equiv \alpha \langle P \rangle$. Periodic boundaries in our DNS imply that $\alpha = V$, (more in (Supplements)). Using this and the fact that the problem has only radial ($r$) dependence, we rewrite (4) as:

$$r^2 \frac{\partial g(r,t)}{\partial t} + \frac{\partial}{\partial r}\left[r^2 \alpha \left(q_i^d + q_i^D\right) + r^2 \langle W_r \rangle g(r,t)\right] = 0, \tag{8}$$

where the content inside [.] gives the total flux. For a system in steady-state, the first term in (8) is zero, and upon integrating with limits $[d, r]$, we have:

$$c_{st}\, r^3 \int d\Omega \int_0^\infty dt_f F(t_f) \int_{d/r}^\infty dR_0\, R_0^2\, g(r R_0)\, f_I(R_0, \mu, t_f)$$

$$+ \quad g(r)\left[r^2 \langle W_r \rangle - A_\tau r^3\right] = -R_c^*, \tag{9}$$

---

[4] Here a positive $q_r^d$ merely reflects a deficit in the inward flux of neighboring particles since we find that $q_r^d + q_r^D$ is always negative.

where we have identified the total flux at contact ($r = d$) as the negative of the (always positive) normalized collision rate
$R_c^* \equiv R_c/(4\pi[N(N-1)/2]/V)$, and comparing with (7), we see that:

$$A_\tau \equiv St \int\limits_{-\infty}^{t} a_{ik}^{(1)} dt' \; + \; St^2 \int\limits_{-\infty}^{t} a_{ki}^{(2)} dt', \tag{10}$$

with the specific values of the $t'$-integrals already given above. For clarity, we reiterate that on the left side of Eq. (9), we have the diffusive flux ($q_r^D$), mean-field flux ($r^2 g(r) \langle W_r \rangle$), drift flux ($q_r^d$); while on the right, the total flux is given in terms of the normalized collision rate ($R_c^*$). We note that this equation embodies the full relationship among RDF, MRV and collision rate.

## 4.2 Ansatz and Accuracy of the Theory

Simple analytical solution to Eq. (9) may be elusive due to its integral nature (a consequence of the non-local diffusive-flux). However, one could gain insights into it and test its accuracy via numerical solutions. To that end, we begin with a simple ansatz for $g(r)$, then we curve-fit the ansatz to the DNS-produced RDF ($g_{\text{DNS}}(r)$). This enables us to, firstly, verify that the ansatz could accurately represent $g_{\text{DNS}}(r)$, and secondly, obtain a "calibrated" ansatz that is a numerically accurate representation of $g_{\text{DNS}}(r)$. We then show that Eq. (9), supplied with the calibrated-ansatz, could numerically predict $\langle W_r \rangle(r)$ (i.e. MRV) that agrees well with the DNS-produced MRV. In short, we will show that given a "correct" $g(r)$, (9) produces the "correct" $\langle W_r \rangle(r)$.

The ansatz has the form $g_a(r) = g_0(r)g_s(r)$, with $g_s(r) = c_0 r^{-c_1}$ i.e. the RDF form for non-colliding particles (Chun et al., 2005) under the same conditions. As a first order analysis, we let $g_0$, which embodies the effects of collision, takes the simplest form that could still capture the main features of the RDFs seen in Fig. 1. Specifically, we let $g_0(r) = c_{00}(r - d_1)^{c_{10}}$, where $c_{00}(r), c_{10}(r)$ are each piecewise constant quantity that switches from its small-$r$ to its large-$r$ values at a crossover-scale $r_c$ (of the order of $d$), i.e. $g_0$ is a two-piece power-law of $r - d_1$. (Note: our earlier finding of $g(r \to d) > 0$ implies that $d_1 < d$.)

From a given DNS-produced RDF ($g_{\text{DNS}}(r)$), we first obtain a calibrated $g_s$ by fitting $c_0 r^{-c_1}$ to $g_{\text{DNS}}(r)$ in the power-law regime $d \ll r \lesssim 10\eta$ (see the red-line in Fig. 1b). Next, we compute the DNS estimate of $g_0$ via $g_0^{\text{DNS}} = g_{\text{DNS}}(r)/g_s(r)$ which is essentially a compensated RDF (see the $\bigcirc$-plot in Fig. 1b). To get a calibrated $g_0$, we then fit the general form of $g_0$ given above to $g_0^{\text{DNS}}$ (see the cyan-line in Fig. 1b, note: each time, two pieces of power-laws are fitted to one $g_0^{\text{DNS}}$, $r_c$ results naturally from the intersection of the two). Fig. 1b shows the calibrated ansatz for the case of $St = 0.054$ and verify its accuracy (the red-line is $g_s(r)$, cyan-line is $g_0(r)$; the black-dashed-line ($g_0 \cdot g_s$) accurately reproduces $g_{\text{DNS}}(r)$). The inset in Fig. 1b shows that $g_0^{\text{DNS}}(r)$ ($= g_{\text{DNS}}(r)/g_s(r)$) is roughly $St$-independent for $St \ll 1$.

Next, we numerically evaluate the integral in the first term of (9). The $St \ll 1$ assumption allows us to approximate $g(r, St)$ inside the integral by its zero-$St$ cousin $g(r, St \to 0)$ (Chun et al., 2005). In practice, we replace $g(r, St)$ with the ansatz fitted to the DNS result of $g(r, St = 0.001)$. Next, we use the DNS data to estimate $A_\tau$, compute $R_c^*$ and $c_{st}$ (for this case, DNS gives $R_c^* = 9.69 \times 10^{-10} \text{dm}^3/\text{s}$; $c_{st} = |c_1|$ as mentioned earlier). Finally we use (9) to predict $\langle W_r \rangle(r)$.

Comparison of the predicted $\langle W_r \rangle(r)$ with the ones obtained directly from the DNS is shown in Fig. 2. The prediction shown was made for the case of $St = 0.054$, to be compared with its DNS counterpart (the $\bigcirc$ symbols). (We also show the DNS result

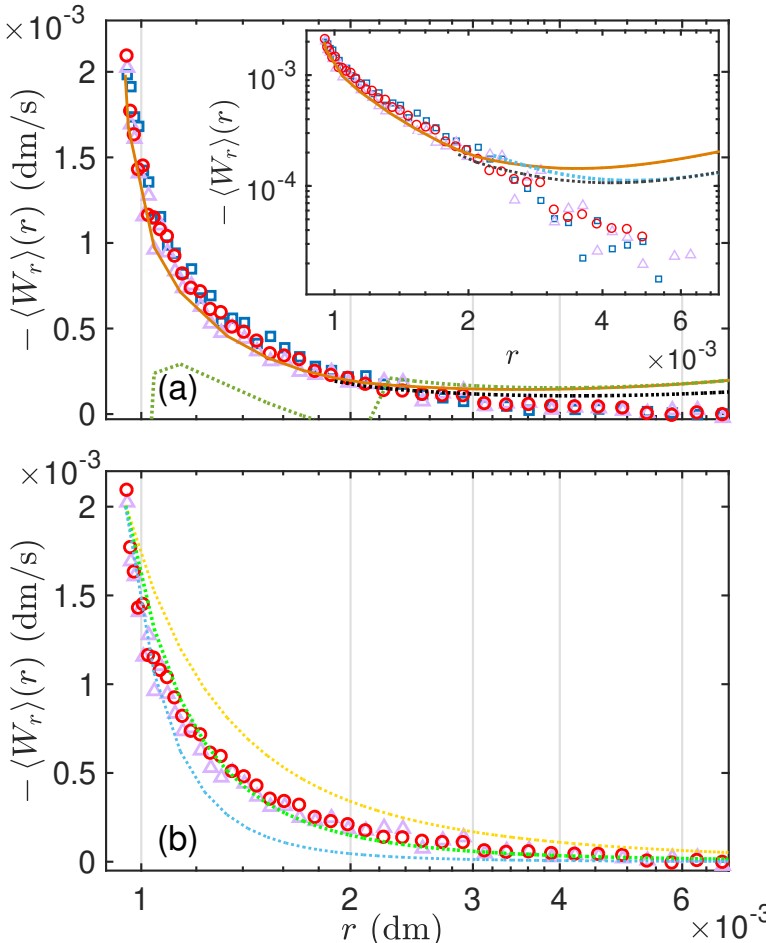

**Figure 2.** Mean radial component of relative velocity (MRV) for particles of specific Stokes numbers and some theoretic-numerical predictions. **a)** Symbols are DNS results with $\triangle$: $St=0.001$; $\bigcirc$: $St=0.054$; $\square$: $St=0.11$. The lines are the numerical predictions by the theories (equation (9) or (13)) using the ansatz (details in text). Orange-line: $\langle W_r \rangle^{\text{theory}}_{r \sim d, St=0.054}$, i.e. the numerical prediction via the integral version of the theory (Eq. (9)) for the small-$r$ regime ($r \sim d$); black-line: $\langle W_r \rangle^{\text{theory}}_{r \gg d, St=0.054}$, same as the previous but for the large-$r$ regime ($r \gg d$); green-line: prediction of the differential version of the theory (Eq. (13)) for the $r \sim d$ regime. **Inset)** A repeat of the main figure in log-log axes. Exception: Cyan-line is the prediction of the differential version of the theory, but for the $r \gg d$ regime. **b)** MRV compared with predictions via the phenomenological model of particle approach angles (Eq. (11) and (12)). DNS results: $\triangle$: $St=0.001$; $\bigcirc$: $St=0.054$. Dotted lines are model predictions of $\langle W_r \rangle_{St=0}$ using (11) and (12) with variance $K$ obtained by matching the model's and DNS's transverse-to-longitudinal ratio of structure functions (TLR) of a certain order (from the top, yellow-line: order 2, green-line: order 4, cyan-line: order 6).

for $St = 0.001$ and $0.11$ to highlight an observation that $\langle W_r \rangle(r)$ is almost $St$-independent in this small-$St$ regime.) We have shown earlier that for $r \sim d$, $A_\tau$ is given by (10). However, as stated earlier, as $r$ increases, the (statistical) asymmetry induced by collision-coagulation gradually becomes subdominant to the isotropy of turbulent-fluctuation. Statistical isotropy implies $a_{ik}^{(1)} = 0$ (Chun et al., 2005), a fact our DNS data confirm. Thus, for $r \gg d$, $A_\tau$ equals the order $St^2$ term in (10), exactly the same as the results of (Chun et al., 2005) for non-colliding particles. For this reason, we show two versions of the prediction: $\langle W_r \rangle_{r \sim d}^{\text{theory}}$ and $\langle W_r \rangle_{r \gg d}^{\text{theory}}$, which are respectively obtained by setting $A_\tau$ to its small-$r$ and large-$r$ limits ($-2.6 \times 10^{-3} s^{-1}$, $7.1 \times 10^{-3} s^{-1}$) respectively. The agreement between DNS and the predictions is noteworthy, especially for small $r$. At $r \approx 2d$, the DNS result shows a weak tendency to first follow the upward trend of $\langle W_r \rangle_{r \sim d}^{\text{theory}}$ and then drops off significantly at $r \gtrsim 2.5d$. The latter is consistent with the fact that $\langle W_r \rangle_{r \gg d}^{\text{theory}}$ is below $\langle W_r \rangle_{r \sim d}^{\text{theory}}$, but the drop is sharper than predicted.

## 4.3  Phenomenological Model of MRV

Alternatively, (9) may be solved for the correct g(r), if $\langle W_r \rangle$ is given. As we are assuming $St \ll 1$, particle velocity statistics may be approximated by their fluid counterparts (Chun et al., 2005), i.e. we may replace $\langle W_r \rangle$ with $\langle W_r \rangle_{St=0}$, the latter being the MRV of fluid particles. Hence, if $\langle W_r \rangle_{St=0}$ is known, it may be used, together with (9), to predict RDF of any finite but small $St$. Fig. 2a shows that $\langle W_r \rangle_{St>0}$ from the DNS do not change significantly for $St \in [0.001, 0.1]$, supporting this approach[5].

Here we provide a simple, first order, model for $\langle W_r \rangle_{St=0}$. We limit ourselves to the regime of small particles ($d \ll \eta$) and anticipate that $\langle W_r \rangle$ is non-trivial (nonzero) only for $r \sim d$, a fact observable in Fig. 2a. We also assume that the relative trajectories of particles are rectilinear at such small scales. The coagulation constraint then implies that: in the rest frame of a particle (call it P1), a second particle nearby must move in such a way that the angle ($\theta$) between its relative velocity and relative position (seen by P1) must satisfy: $\sin^{-1}(d/r) \leq \theta \leq \pi$, under the convention of $\sin^{-1}(x) \in [-\frac{\pi}{2}, \frac{\pi}{2}]$, (more in (Supplements)). We can thus write (by treating negative and positive $w_r$ separately, applying the K41-theory (Kolmogorov, 1941) and the bounds on $\theta$, details in (Supplements)), for $St \ll 1$, that:

$$\langle W_r \rangle \equiv \langle w_r \rangle_* = p_- \langle w_r \, | \, w_r < 0 \rangle_* + p_+ \langle w_r \, | \, w_r \geq 0 \rangle_*$$

$$\approx -p_- \xi_- r \; + \; p_+ \xi_+ r \left[ 1 + \frac{\int_{\theta_m}^0 P_\theta^+(\theta') \cos(\theta') d\theta'}{\int_0^{\frac{\pi}{2}} P_\theta^+(\theta') \cos(\theta') d\theta'} \right], \quad (11)$$

where $\langle . \rangle_*$ denotes averaging over particle pairs, $p_+$ ($p_-$) is the probability of a realization of $w_r$ being positive (negative), and $P_\theta^+$ is a conditional PDF such that $P_\theta^+ \equiv P(\theta \, | \, w_r \geq 0) \equiv P(\theta \, | \, \theta \in [0, \frac{\pi}{2}])$, $\theta_m$ is the lower bound of $\theta$ described above. For a first order account, we neglect skewness in the distribution of particle relative velocities and set $p_\pm = 0.5$. Following (Kolmogorov, 1941), we have set $\langle w_r \, | \, w_r < 0 \rangle_* = \xi_- r$, where $\xi_\pm = C_s \sqrt{\varepsilon/(15\nu)}$, ($C_s$ is a Kolmogorov constant, we found $C_s = 0.76$ by matching $\xi_- r$ to the first-order fluid velocity structure-function from the DNS).

---

[5]This is true in the relatively idealized system simulated, but may not apply to the general problem that includes other effects

A simple phenomenological model for $P(\theta)$ may be constructed using the (statistical) central-limit-theorem by assuming that the angle of approach $\theta$ at any time is the sum of many random-incremental rotations in the past, thus we write:

$$P(\theta) = N \exp[K \cos(\theta - \mu_\theta)] \sin(\theta), \tag{12}$$

where $N\exp[...]$ is the circular normal distribution, i.e. analog of Gaussian distribution for angular data; $\sin(\theta)$ results from integration over azimuthal angles ($\phi$). We set $\mu_\theta = \frac{\pi}{2}$ (neglect skewness in fluid's relative velocity PDF) and obtain $K$ by matching the transverse to longitudinal ratio of structure functions (TLR) of the particle relative velocities with the ones via the DNS data; $N$ is determined via normalization of $P(\theta)$. Fig. 2b shows the $\langle w_r \rangle_*$ derived via (11) and (12), using $K$ calibrated with TLR of 2nd, 4th, 6th order structure functions respectively. The results have correct qualitative trend of vanishing values at large $r$ that increases sharply as $r$ approach $d$, with the 4th-order's result giving the best agreement with DNS. Currently we have not a satisfactory rationale to single out the 4th-order. The TLR of different orders give differing results may imply that our first-order model may be incomplete, possibly due to over-simplification in (12) or to the inaccuracy of the rectilinear assumption ($d/\eta$ in the DNS may be insufficiently small).

## 4.4 Differential Version of the Theory, Its Validity and Solution

We now discuss an important but precarious theoretical issue. Chun et al. (2005) clearly showed that the non-local diffusion ($q_r^D$) may be converted, from its general integral form, into a differential version only when the underlying RDF is a simple power-law. However, Lu et al. (2010) and Yavuz et al. (2018), working in two very different scenarios, found that their predictions using the differential form of the theory agree well with experiments, even when the RDFs involved was clearly not power-laws. We shall attempt to remedy this apparent paradox in future work. To examine how well this albeit unjustified method works here, we recast (9) into its differential form (Chun et al., 2005):

$$-\tau_\eta^{-1} B_{nl} r^4 \frac{\partial g}{\partial r} + g(r) \left[ r^2 \langle W_r \rangle - A_\tau r^3 \right] = -R_c^*, \tag{13}$$

where $B_{nl} = 0.0397$ (this value is computed from our DNS, $B_{nl}$ is expected to depend on flow characteristics e.g. $R_\lambda$ and $\tau_\eta$ (more in (Supplements)). Using (13), the same $g_s g_0$ ansatz, we make another prediction for $\langle W_r \rangle_{St=0.054}$, which is plotted in Fig. 2a (green dash-line). This prediction is far from the DNS at $r \sim d$ but perform as well as the integral version at $r \gg d$ (the jump in the curve is just an artifact from the kink in the ansatz).

One advantage of (13) is that it admits a general solution, which we now give, assuming $\langle W_r \rangle$ is given by (11) & (12):

$$g(r) = \frac{1}{\beta(r)} \left[ \int \beta(r) q(r) dr + C \right], \tag{14}$$

with $q(r) = R_c^* \tau_\eta / (B_{nl} r^4)$; $\beta(r) = \exp \left[ \int p(r) dr \right]$; $p(r) = [A_\tau r - \langle w_r \rangle_*] \tau_\eta / (B_{nl} r^2)$, (more in (Supplements)).

## 4.5 Effects of Gravity and Other Limitations

Thus far, we have not considered the effects of gravity on the particles. Here we provide a glimpse on the role of gravity (a detailed analysis is beyond the scope of the present study). In keeping with the scope of current work, we restrict ourselves to

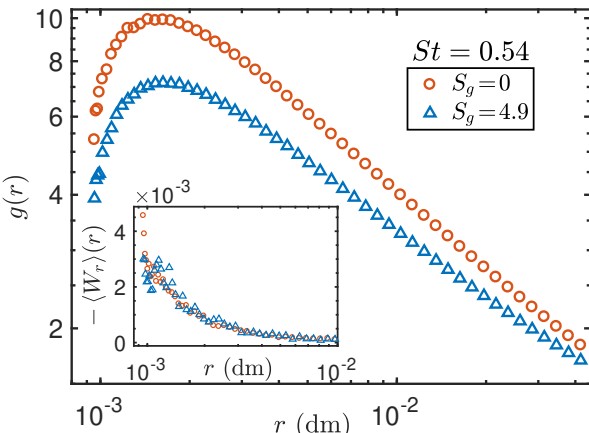

**Figure 3.** RDFs of particles ($St = 0.54$) subject to action of turbulence, collision-coagulation with and without gravity. Circles: $S_g = 0$ (zero gravity); triangles: $S_g = 4.9$ (nonzero gravity). The latter shows a reduced slope in the power-law regime, while the shape of the two curves largely similar in the collision regime ($r \sim d$). **Inset)** MRVs of the same cases as in the main figure. Gravity weakens the MRV of the particles.

the case of monodisperse particle only. For this, we rerun the DNS cases of $St = 0.054$ and $0.54$ with gravity (to be compared with the zero-gravity case). The new particle advection equation is: $d\mathbf{v}/dt = (\mathbf{u}-\mathbf{v})/\tau_p + \mathbf{g}$ (all other details of the DNS remain unchanged). We choose to have the particle settling parameter $S_g \equiv \tau_p g / u_\eta$ (where $u_\eta$ is the Kolmogorov velocity scale) be in the range $O(0.1) - O(1)$ (this is achieved by letting $|\mathbf{g}| = 10\,\mathrm{dm/s^2}$). As a result, the range of $S_g$ and $St$ explored here are well aligned with measured values in natural clouds (Siebert et al., 2010). For the case of $St = 0.054$ ($S_g = 0.49$), we find no discernible difference for both RDF and MRV between the "with gravity" and zero-gravity results (corresponding figures in (Supplements)). For the $St = 0.54$ ($S_g = 4.9$) case, Fig. 3 shows the effects of gravity on the RDF and MRV. We see that the slope (exponent of $g(r)$ in the range $d \ll r < 20\eta$) of the RDF in the gravitational case is reduced by about 15% compared to the zero-gravity case ($S_g = 0$). However, the shape of the RDF in the collision regime ($r \sim d$) is approximately preserved, suggesting that a construct of the form $g_{\mathrm{collision}} \times g_{\mathrm{gravity}}$ may be a good first order model for the full RDF (close examination of the compensated RDFs gives substantial support for this idea, details in (Supplements)). These observations imply that as $S_g$ increase from $O(0.1)$ to $O(1)$, the effects of gravity on RDF grow from negligible to significant but not dominant, the main effect is the reduction of the exponent while the collision related "modulation" ($g_{\mathrm{collision}}$) remains largely intact. The inset of Fig. 3 shows that the MRV is also weakened by gravity, albeit the statistical noise limits the strength of this conclusion. Lastly, It is worth noting that in the complimentary DNS by Woittiez et al. (2009) that included gravity but not actual collisions, much stronger gravitational effect was found on the statistics of bidisperse particles relative to the monodisperse case.

As mentioned, the fundamental focus of our work precludes the DNS and theory from considering a number of complexities relevant to some applications. As a result, this limits the direct quantitative applicability of our results to some realistic problems

(e.g. in clouds). Besides gravity, another neglected factor is the hydrodynamic inter-particle-force (HDI). Recent works e.g. (Yavuz et al., 2018; Bragg et al., 2022) found that HDI also has strong impact on RDF for $r \sim d$. For monodisperse particles with small to moderate $St$, HDI is expected to be more important than gravity. While we expect that HDI should not alter the qualitative trend that $g(r)$ should fall towards a small value at $r \to d$ (the same applies to the observed trend of MRV), it is likely that HDI and collision would affect RDF and MRV in a coupled manner.

Also neglected is the influence of temperature, humidity and vapor-liquid phase transition which are important in the atmospheric clouds. These factors have substantial impact on the polydispersity of small droplets (see e.g. (Kumar et al., 2012, 2014)). However, for monodisperse statistics considered here, they are likely to play minor role (they will be more important when future works consider the full polydisperse problem).

One limitation of the theory stems from the assumption of $St \ll 1$ and its corollary that particle velocity statistics in this regime are $St$-independent (Chun et al., 2005), which limit the theory's applicability to real systems. This implies that MRV should be $St$-independent in this regime. Our DNS results (spanning two orders of magnitude in $St$) shown in Fig. 2 give some support to the latter. However, unlike the theoretical prediction for MRV of case $St = 0.054$ (Fig. 2), we have found that the prediction for $St = 0.11$ is discernibly below the DNS result (figure in (Supplements)). This could be due to the finite $St$ effect not captured by the theory or other reasons (details in (Supplements)). Hence, a finite $St$ extension of the theory is desirable to improve its applicability to real systems.

## 5    Conclusions

To conclude, we observed that collision strongly affects the RDF and MRV and imposes strong coupling between them[6]. This challenges the efficacy of a "separation paradigm" and suggests that results from any studies that preclude particle collision has limited relevance for predicting collision statistics. We have presented a theory for particle collision-coagulation in turbulence (based on a Fokker-Planck framework) that explains the above observations and verified its accuracy by showing that $\langle W_r \rangle$ could be accurately predicted using a sufficiently accurate RDF. The theory account for the full collision-coagulation rate which includes contributions from mean-field and fluctuations; and as such, our work complements and completes earlier mean-field theories (Saffman and Turner, 1956; Sundaram and Collins, 1997). We showed that a simple model of particle approach-angles could capture the main features of $\langle W_r \rangle$ and used it to derive a general solution for RDF from the differential version of the theory. We uncovered a possible paradox regarding the past empirical successes of the differential drift-diffusion equation (see Sec. 4.4). Further shape-preserving reduction of the RDF and MRV were observed when gravitational settling parameter ($S_g$) is of order $O(1)$. Our findings provide new perspectives of particle collision and its relation with clustering and relative motion, which have implications for atmospheric clouds or generally to systems involving colliding particles in unsteady flows.

---

[6]This statement also holds for other types of collisional outcomes (not only for collision-coagulation), but the details of the specific outcomes should be different from the current case.

*Author contributions.* All the authors made significant contributions to this work.

*Competing interests.* The authors declare no competing financial interests.

*Acknowledgements.* This work was supported by the National Natural Science Foundation of China (Grant 11872382) and by the Thousand Young Talent Program of China. We thank Jialei Song for helps. We thank Wai Chi Cheng, Jianhua Lv, Liubin Pan, Raymond A. Shaw for discussion and suggestions.

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
