# Peer review of "Intricate Relations Among Particle Collision, Relative Motion and Clustering in Turbulent Clouds: Computational Observation and Theory."

_Atmospheric Chemistry and Physics, 2021_

## Referee Comment (RC1)

**Review**

Title: Nontrivial Relations Among Particle Collision, Relative Motion and Clustering in Turbulent Clouds: Computational Observation and Theory

Authors: Ewe-Wei Saw and Xiaohui Meng

**Summary**

The authors presented relationship among particle collision radial distribution function (RDF) and mean relative velocity (MRV). They compared the results from Direct Numerical simulation (DNS) and theory proposed by them. The results show that collision is affects the RDF and MRV and imposes strong coupling between them.

The work done in the manuscript is novel. However, authors have to provide more details of the numerical simulations before it can be considered for publication in the journal. The major comments are provided below.

Major Comments

1. More information about the DNS has to be provided in the introduction section. In particular, the reference should be given for DNS done by others using similar condition of turbulence and numerical schemes. See the references Kumar et al. (2012,2014) and Thomas et al. (2020).
2. Section 2:
   (i) Some references are required for the DNS set up.
   (ii) More details about particle characteristics such as initial size distribution. The number density, temperature and humidity used for the simulation.
3. Section 4 (or 2): Simulation details:
   (i) Is there and phase change considered for the particles?

4. The article starts with "considering turbulent clouds containing small heavy particles", however, gravitational acceleration is not considered in particles velocity equation (Page-3, line 72, equation number is not given). Justification for this, given by the authors, is not strong enough. The entire study was done to investigate the effect of particle collision on particle relative motion. Hence in this scenario, effect of gravity should be considered in droplets velocity equation.
5. Definition of $\tau_p$ (page 3, line 73) is not mentioned in the article.
6. Detail information about particles number concentration at the time of injection, surrounding temperature and relative humidity of particles is also missing in the article.
7. Justification behind the consideration of certain values of Stokes number (St = 0.22, 0.54, 0.054, 0.001) (page-5 line113) is missing in the article. While presenting an accuracy of a theory, a large number of simulations are needed at all possible complex scenarios. I found it missing in authors' present work.

8. As per my understanding, authors considered particles of size 94.9 micrometre (0.000949 dm) or bigger. If so then mention the DNS domain size and each grid box length. Also, at what stage, the reverse effect of particle collision was considered is missing in this article.

9. The statement made by authors in section 4.3 (page 10, line 211) is not justified. The numerical simulation is done at a very basic scenario, which may lead to an insignificant impact of Stokes numbers on $W_r$, calculated using DNS.

10. In my opinion, authors need to work more to justify their work and give a more clear explanation about the results.

At this stage, the article is not ready for the publication and requires major changes.

References:

Kumar, B., Schumacher, J., and Shaw, R. A.: Lagrangian Mixing Dynamics at the Cloudy-Clear Air Interface, J. Atmos. Sci., 71, 2564–2580, https://doi.org/10.1175/JAS-D-13-0294.1, 2014.

Kumar, B., Janetzko, F., Schumacher, J., and Shaw, R. A.: Extreme responses of a coupled scalar–particle system during turbulent mixing, New J. Phys., 14, 115020, https://doi.org/10.1088/1367-2630/14/11/115020, 2012.

Thomas, L., Grabowski, W. W., and Kumar, B.: Diffusional growth of cloud droplets in homogeneous isotropic turbulence: DNS, scaled-up DNS, and stochastic model, Atmos. Chem. Phys., 20, 9087–9100, https://doi.org/10.5194/acp-20-9087-2020, 2020.

---

## Author Comment (AC1)

**Respond to Reviewer 1.**

The authors presented relationship among particle collision radial distribution function (RDF) and mean relative velocity (MRV). They compared the results from Direct Numerical simulation (DNS) and theory proposed by them. The results show that collision is affects the RDF and MRV and imposes strong coupling between them.

The work done in the manuscript is novel. However, authors have to provide more details of the numerical simulations before it can be considered for publication in the journal. The major comments are provided below.

>> We thank the reviewer for appreciating the novelty of our work and for his/her comments and suggestions, we find these comments rather helpful in improving the quality of the manuscript.

Major Comments

1.  More information about the DNS has to be provided in the introduction section. In particular, the reference should be given for DNS done by others using similar condition of turbulence and numerical schemes. See the references Kumar et al. (2012,2014) and Thomas et al. (2020).

>> We have added more information about the DNS and more citations to works on settings similar to our work. We should also mention that the original manuscript had already cited a number of early papers involving DNS under similar setting such as Chun et al. (2005), Sundaram & Collins (1997), Salazar et al. (2008), Saw et al. (2012b, 2014), Voßkuhle et al. (2013), Wang et al. (2008), Bec et al. (2007). We have now also cited some of the references mentioned above by the reviewer where suitable.

2.  Section 2:
    (i) Some references are required for the DNS set up.
    (ii) More details about particle characteristics such as initial size distribution. The number density, temperature and humidity used for the simulation.

>> (i) More relevant citations are now included. We should also point out that a significant number of citation was already given for the set-up of our DNS, for instance: we have cite Rogallo (1981), Pope (2000) and Mortensen (2016) for the standard pseudospectra method and the general, standard, DNS of this type; we have cited Ireland (2013) for the novel time-stepping method used for advancing the particles; We have cite Rogallo (1981) for the standard 2/3-dealiasing rule employed. Now, we have also added citations for the particle advection equation and the collision-processing procedures among other things. (ii)These details are now described in the manuscript (in particular in Section 2). We have also added a dedicated section in the Supplement with further details on the DNS.

3.  Section 4 (or 2): Simulation details:

    (i) Is there and phase change considered for the particles?

>> No, this is not considered. We have now made this point clear in Section 2 and discuss the limitations due to this neglect.

4. The article starts with "considering turbulent clouds containing small heavy particles", however, gravitational acceleration is not considered in particles velocity equation (Page- 3, line 72, equation number is not given). Justification for this, given by the authors, is not strong enough. The entire study was done to investigate the effect of particle collision on particle relative motion. Hence in this scenario, effect of gravity should be considered in droplets velocity equation.

>> As we have now explain in the introductory section of the manuscript: "the focus of our work is on the fundamental relationship between collision, RDF and MRV, and to highlight differences from the case with non-colliding particles (Chun et al., 2005). To that end, we have designed the DNS to have an idealized setup similar what was done in (Chun et al., 2005), which would allow us to identify without doubt the effects of particle collision-coagulation." Also: "This approach certainly limits the direct applicability of our results to real problems in atmospheric clouds, these limitations will be discussed in Sec. 4.5."

Another goal of this study is to probe the accuracy of the drift-diffusion theory. In our view, considering the focus and goals of our present work, it is beneficial to keep the DNS setting idealized (and in the regime relevant for the theory) for the sake of clarity when interpreting results. Thus, we choose at this stage to not include various complexities that, although important for the ultimate application to real atmospheric clouds, would distracts the reader from the simple fundamental message derivable from our work, i.e. 1) Collision will fundamentally change the form and relations of RDF and MRV;  2) RDF and MRV are strongly coupled (in general); 3) RDF is not zero at r=d despite appearance; 4) the drift-diffusion theory is, in principle, extendable to scenario beyond non-colliding particles. 5) Simple phenomenology could reproduce the main features of MRV (at least in the regime considered). 6) The differential version of the theory should not have been accurate in some commonly encountered (studied) scenario, thus, pressing theoretical works is needed to understand how it was able to produce reasonable results.

We have now clarified these points in the manuscript (in Sec. 1 and Sec. 2).

However, we appreciate the point of view of the reviewer on the importance of gravitational effects. To that end, we have dedicated a section in the later part of the manuscript to briefly survey the effects of gravitational body force on the main qualitative findings/conclusion of our present work. A detailed quantitative analysis is beyond the scope of this manuscript and should be a subject of future studies.

To address the general concern of the reviewer on the relevance to realistic clouds, we have added a section that clarify the limitations of the present study. In short, our results are not intended to be directly applicable to real applications but are intended as a tool to explore several fundamental questions related to the problem. It is hoped that these fundamental understandings will inform future, more realistic, accounts of the complex problem.

5. Definition of $\tau_p$ (page 3, line 73) is not mentioned in the article.

>> We thank the reviewer for pointing out this error. A definition is now given.

6. Detail information about particles number concentration at the time of injection,

surrounding temperature and relative humidity of particles is also missing in the article.

>> Question regarding temperature and humidity is already addressed in point 2 and 4 above. The initial number of particles is now reported in Sec. 2. Also, more technical details about the simulation are now given in a dedicated section in the Supplement.

7. Justification behind the consideration of certain values of Stokes number (St = 0.22, 0.54, 0.054, 0.001) (page-5 line113) is missing in the article. While presenting an accuracy of a theory, a large number of simulations are needed at all possible complex scenarios. I found it missing in authors' present work.

>> A motivation for choice of St is now given in the second paragraph of Sec. 4.

>> As the theory is derived based on the assumption of St<<1. We are exploring the Stokes number values close to this regime only, with the exception of St=0.54, which is included to show that the general trend we observed (on the DNS produced RDF and MRV) extends beyond the St<<1 regime. An extension of the theory to finite Stokes number is definitely of great interest (even without including gravity and other inter-particle forces) and has been pursued for decades now, it is hoped that our results could shed light on this endeavor.

8. As per my understanding, authors considered particles of size 94.9 micrometre (0.000949 dm) or bigger. If so then mention the DNS domain size and each grid box length. Also, at what stage, the reverse effect of particle collision was considered is missing in this article.

>> Domain size (of the simulation cube) was already given in the original manuscript in Table 1 (i.e. 2*pi dm on each side). The number of grid points (i.e. 256^3) was also already given in Sec. 2 (just after Eq. 1). The reverse effect is always present, as a consequence of collision-coagulation, we do not directly control or impose this.

9. The statement made by authors in section 4.3 (page 10, line 211) is not justified. The numerical simulation is done at a very basic scenario, which may lead to an insignificant impact of Stokes numbers on $Wr$, calculated using DNS.

>> We understand that here the reviewer means that since our DNS setup is fairly idealized, we may not be able to capture any extra impact (on the MRV) caused by forces omitted in the DNS (e.g. gravity), thus the said statement would not hold for the general, real, system. We basically agree with that point of view, and thus we have added a footnote to clarify this limitation.

10. In my opinion, authors need to work more to justify their work and give a more clear explanation about the results.

At this stage, the article is not ready for the publication and requires major changes.

>> We thank the reviewer for these comments, and we have carefully edited the text, added sections and scientific contents to address these comments (details above). In our view, the updated manuscript should satisfy the required standards. We reiterate that this study focuses on the fundamental aspects of the problem (details in respond to point 4 above and the manuscript) and it is in the interest of scientific clarity that the DNS is done in a relatively

idealized setting. Also, it is our opinion that the fundamental findings our manuscript conveys merit rapid publication so that the community could benefit from them, before a more detailed quantitative investigation involving more complexities could be done.

References:

Kumar, B., Schumacher, J., and Shaw, R. A.: Lagrangian Mixing Dynamics at the Cloudy-Clear Air Interface, J. Atmos. Sci., 71, 2564–2580, https://doi.org/10.1175/JAS-D-13-0294.1, 2014.

Kumar, B., Janetzko, F., Schumacher, J., and Shaw, R. A.: Extreme responses of a coupled scalar–particle system during turbulent mixing, New J. Phys., 14, 115020, https://doi.org/10.1088/1367-2630/14/11/115020, 2012.

Thomas, L., Grabowski, W. W., and Kumar, B.: Diffusional growth of cloud droplets in homogeneous isotropic turbulence: DNS, scaled-up DNS, and stochastic model, Atmos. Chem. Phys., 20, 9087–9100, https://doi.org/10.5194/acp-20-9087-2020, 2020.

---

## Author Comment (AC2)

Response to Anonymous Referee #2

Referee comment on "Nontrivial Relations Among Particle Collision, Relative Motion and Clustering in Turbulent Clouds: Computational Observation and Theory" by Ewe-Wei Saw and Xiaohui Meng, Atmos. Chem. Phys. Discuss., https://doi.org/10.5194/acp-2021-844-RC2, 2021

Review of "Nontrivial Relations Among Particle Collision, Relative Motion and Clustering in Turbulent Clouds: Computational Observation and Theory" by Saw and Meng

This article reports on theoretical modeling of the clustering and relative velocity of inertial particles in homogeneous, isotropic turbulence. The crucial difference between this work and much of the inertial particle work in the literature, is that particles are allowed to collide and coagulate (coalesce). The theoretical model addresses the now-25-year-old suggestion that collision rate can be expressed in a generalized form, being the product of the radial distribution function (which represents the tendency of inertial particles to 'cluster' in turbulence) and the the mean radial velocity (which represents the tendency of inertial particles to have an average inward relative velocity).

The theoretical effort rests on the output from direct numerical simulation (DNS) of the Navier-Stokes equations for a triply-periodic box of turbulence containing inertial particles with small Stokes numbers. The key, novel findings from the paper are: 1) accounting for collisions leads to an extreme drop off in the radial distribution function at small droplet separation distances, on the order of the droplet diameter (the collision scale); 2) an apparently empirical power-law fit to this drop off in the radial distribution function, together with assumptions about randomness and linearity of droplet relative trajectories, leads to expressions for the mean relative velocity that match the DNS results with reasonable success.

Although there have been suggestions of finding 1 in the previous literature, the extension of drift- diffusion theory (Fokker-Planck method used by Chun et al.) to account for collisions provides important new insight into the fundamental link between the two contributions to the generalized collision rate. In short, the notion of independent contributions of radial distribution function and mean relative velocity is invalid. The findings presented here show how they are related to each other, which should lead to simplification of the general collision problem. Furthermore, the finding that a simple model of particle approach angles can successfully describe the behavior of the mean relative velocity is important.

The paper is technically sound, as far as I am able to follow the derivations, and the central findings are of relevance to the cloud droplet collision problem. Recommendations to improve the clarity and interpretation of the paper are given below.

>> We thank the referee for these rather positive comments. We find pleasure in knowing that the referee has a deep understanding of our work and appreciates the potential impact of some of our findings.

Grammar check: The paper is written in a reasonably clear way, but there are many grammatical errors, especially agreement of case. Even an automated corrector like Grammarly would significantly improve this.

>> We thank the referee for this suggestion, we have use Grammarly as suggested & have corrected the errors found.

Title: I don't find the "Nontrivial Relations" in the title very compelling. It's the authors' call on this, but I would recommend changing it so that the title conveys a clear physical meaning. For example, "Strong coupling of particle collision, relative motion and clustering in turbulent clouds".

>> We appreciate this view by the referee. We also see the benefits of highlighting the most important finding of the paper in the title (like the suggestion). But unfortunately, our view is that the paper conveys more than just the "strong coupling". And one of our intended messages is that there are still more mysteries to be solved, e.g. the apparent paradox pertaining to the historic accuracy of the differential version of the theory, the role of other forces and so on. It is our subjective feeling that while we have found some solid answers in this work, we are unleashing more interesting questions with which we are still toying/struggling with to date. In that spirit, we thought "nontrivial" would be closer to what we wish to convey, as we feel the ultimate answer is yet to come. However, we might agree with the referee that it may not be the best choice of word, so we shall change it to "Intricate Relations" instead.

Abstract, first sentence: The meaning of "the reverse effect" is unclear.

>> We have remove the word "reverse" from this sentence to avoid confusion.

Abstract, second sentence: This is vague, please specify that DNS for St<1 and St<<1 are conducted (i.e., define "various cases").

>> We have now specified the regime of interest and the range of St covered in the abstract.

Abstract, lines 9-10: It's strange to read that the "theory accurately accounts for the DNS results" but then that a phenomenological model is being developed. Perhaps this can be clarified.

>> This is now further clarified in the abstract.

Lines 25-29: It's confusing to discuss RDF and MRV before they are formally defined. I recommend keeping the discussion general here, and not referring specifically to terms that will be confusing.

>> Modification has been made to be in line with this comment. The discussion is now general.

Line 34: MRV as defined is averaged over all *approaching* particle pairs, correct?

>> That is correct. Since this conditioning is already clear in the mathematical symbol used, we have revise the text to state only the general phrase "averaged over particle pairs" which is meant to clarify the meaning of the symbol $\langle \dots \rangle_*$. In relation to this, one should note that the condition of "approaching particle pairs" is only needed when one is working under the "separation paradigm" i.e. considering ghost-particles that never collide, because in that case the unconditional MRV is nominally zero.

Line 39: It would be more accurate to state that the other theories are similar, rather than equivalent.

>> We have incorporated this change.

Lines 83-85: How is the coagulation growth balanced to achieve steady state? Is there a loss term for the particles that are growing by coagulation?

>> We focus on the statistics of "monomers" only (i.e. the particle of the smallest size that we have introduced into the system initially and later replenished at a constant rate close to the monomer-monomer collision rate). In this sense, monomers are naturally lost (from our consideration) once they collide and become larger particles (which we ignore). Hence, the lost rate of monomers equals monomer-monomer collision rate. Also, particles that become much larger ($St > 22$) is removed from the DNS. We now describe these details in a dedicated section in the Supplement.

Table 1: The use of dm is strange. Is there a compelling reason for this? The authors can do what they prefer, but using m or cm would be more typical. Also, in the caption, no units are given for $d*$.

>> We prefer to keep "dm" as the spatial unit as it is not scientifically incorrect. Historically, our DNS was carried out without any specific units, this is not an uncommon practice among DNS workers. Ultimately one should look at normalized or dimensionless parameters when interpreting or comparing results. Units were added due to reviewer request. However, we have a vast amount internal notes without explicit units. The choice of dm allows us to keep drawing from these notes without risk of errors. The missing units of $d*$ is now added.

Line 89: "closely tied to the fluid velocity..."

>> We have incorporated this change.

Lines 105 and 107: I recommend giving equation numbers for $q_i^d$ and $q_i^D$ so they can be referred to later. Specifically, there is no need to squeeze the definitions into lines 145-146, instead they can be cited.

>> We have incorporated this change.

Lines 112-113: I was confused by the Stokes numbers because of the way the sentence is written (it looks like the RDFs are for particles of different Stokes numbers). Reformat to make it clear.

>> Changes have been made to prevent this confusion.

Lines 130-133: It would help to move this sentence to earlier in the section where the figure is first introduced.

>> We prefer to highlight the most important (new) findings by discussing them first. The observation those lines refer to, in our view, is just a confirmation of very well-established result. However, we will now mention/hint right after the figure's debut, that power-laws are observed.

Figure 1 caption: Is there a reason for defining $d_1$ as $0.99d$?

>> The ultimate reason is an empirical one. This results from the fit of the two-piece-power-law to the DNS results. We have now clarified this in the manuscript.

Equation 5: This is a repeat of equation 4, so it can be deleted. My sense is that lines 139-144 and the accompanying footnote can be moved to the previous section and the result would be much clearer.

>> We have now significantly reduced these redundancies.

Line 148: The meaning of "nontrivial effects" is unclear.

>> This phrase is unnecessary, we have now revised (simplified) the sentence.

Line 151: Can a physical interpretation of the two $A_{ik}$ terms be provided?

>> We now given some physical interpretation for these quantities (now denoted as $a_{ik}$ to avoid confusion).

Lines 155-156: Is there a physical interpretation for why the drift flux is positive for St less than order 0.01?

>> We currently do not have a satisfactory interpretation for this. We merely added a footnote explaining that: Here a positive $q_r^d$ merely reflects a deficit in the inward flux of neighboring particles since we find that $q_r^d + q_r^D$ is always negative. In other words, it does not mean that particles are actually drifting away from each other.

Line 162: Define \Omega.

>> This is now corrected.

Line 166: It is stated that the addition of the factor $c_{st}$ is 'crucial'... please provide an interpretation of what this represents. Is it purely empirical in its motivation?

>> Yes, it can be understood as an empirical correction. A further motivation of the choice of word "crucial" is now given in a footnote in the section in the Supplement which (in the original draft of this paper) already contained the derivation that shows that it is crucially needed in order for the theory (integral version) to be consistent with empirical results of e.g. Lu (2010), Yavuz (2018) and DNS results.

Equation 8: Formatting is strange.

>> This is to anticipate that the equation will need to fit in the two-column format of ACP. So, this shall remain.

Line 187: The form of the 'ansatz' suggests that there is statistical independence between $g_0$ and $g_s$. Please discuss.

>> We have added "The inset in Fig.2b shows that $g_0(r)$ ($= g_{\text{DNS}}(r)/g_s(r)$) is roughly St-independent in this regime.". We cannot claim that this is generally true (this shall be a subject of future works).

Line 193: Should it be St=0.054?

>> Yes. Corrected.

Figure 1b: The compensated $g(r)$ for $r>r_c$ is flat, so does that imply that $c_{10}$ is 0 in for that part of the piecewise function? If so, this should be stated, e.g., that the form on line 189 only applies in a nontrivial way to the r.

>>Yes, g_{DNS}/g_s which is basically g_0 is nearly flat in that regime and g_0 is nearly equals to constant "1". But we do not make this assumption, we merely do the curve fitting and found that (as reported in the caption). $g\_0 = 1.00(r-d_1)^{-2\times10^{-4}}$ in that regime. Then, we use this result, as is, in the numerical calculation of MRV later. We do not understand the meaning of "that the form on line 189 only applies in a nontrivial way to the r". The paragraphs concerned is now heavily rewritten to make it more clear, extra line is also shown in the figure. We believe these should have addressed the concerns the referee has on this point.

Figure 2a: The labels for the x-axis appear to be cut off. Also, the lines are difficult to see, the orange line isn't visible for small r.

>> The two panels share the same labels and ticks, so the labels of the top panel is supposed to be hidden, this is the case when the paper is compiled into a two column ACP format, but however, when compiled in the current preprint format, it looks like this. Please ignore them. We have also made changes to hide them better. We have now made changes to make the lines more visible. The orange line is now a solid line. We have removed the black-line from the small-r regime, since it is a prediction that is only valid for large-r, we thus show it in that regime only. These changes should reduce clutter and make the lines more visible.

>> We also corrected an error on the green-line i.e. prediction from the differential version of the theory (in the past we have shown a wrong version that does not correspond to the original CK theory. It was a generalized version that incorporates a modification to improve the accuracy over the original CK version— such modification is outside the scope of this paper).

Figure 2: Can an interpretation be provided for why there is so little difference for the large and small-r limits of the integral in Equation 8?

>> This is related to our speculation (explained in the Supplement) that there may be another factor of ~1/2 missing in the definition of the diffusive flux. The related discussion and figure can be found in the Supplement, where we show that adding this empirically inspired factor changes the MRV prediction especially in the large-r regime, which make it much closer to DNS results. Without this factor, as we can see in Fig. 2a of the main text of the manuscript, the predictions deviate significantly from the DNS results in the large-r regime. However, since this is somewhat speculative, we choose not to include this discussion in the main manuscript (it will be a subject of future works).

Line 205: It is stated that the agreement between DNS and predictions is 'remarkable', but is it really? The theoretical treatment is pretty dense, so it's difficult to come away with a clear understanding of how many adjustable parameters are included. The more adjustable parameters, the less remarkable agreement would appear to be. Some discussion of this point would be helpful in guiding the reader and helping to understand just how remarkable the finding is.

>> In retrospect, this choice of word has more to do with our initial excitement over the results. We now revise it to "the agreement … is noteworthy". We hope the rewriting of the paragraphs relating to the numerical solution to the theory (mentioned earlier) should make the "theoretical treatment" appears less dense.

Lines 256-257: The breakdown of the separation paradigm is a significant finding. It implies that the problem is somehow simpler that a product of independent contributions

to the clustering and the approach velocity. How reasonable is it to think about this problem now being reduced in complexity?

>> In the way that the referee frames it, yes it does appear simpler conceptually. But in reference to past works, including Chun et al. (2005), the idea was that it would be easier, if not simpler, to solve one problem at time (separately), and later combine the results. For example, Chun et al. assume that the velocity statistics is given by the fluid statistics then move on to solve for the RDF. The referee raises an interesting point, one that have made us ponder, and we shall continue to think about this in the future. However, at this point, before further investigations, we are unable to formulate very conclusive remarks on this. Perhaps one unintended value of our work is the suggestion that only the velocity statistics (MRV and beyond) matter, the RDF is just a manifestation of the action of the velocities (i.e. particles are denser somewhere because the velocities "prefer" to bring them there). But as we said, we have not fully convinced ourselves on this point and thus prefer not to discuss this in the current manuscript.

Section 5: It would be useful to discuss the potential role of micro-hydrodynamic interactions between particles. Those have not been included, but one would assume that they will become significant at nearly the same scales as the currently observed adjustments to the RDF and MRV. One possible implication is that collision-efficiency effects will influence the RDF and MRV in a coupled fashion, rather than independently.

>> This is now discussed in Section 4.5. A discussion of the effects of gravity could also be found there.

Line 263: The statement "we uncovered the unexplained accuracy of the differential drift- diffusion equation" is vague, so please clarify.

>> The sentence has been rewritten.

Supplement: The title is incorrect... should be updated to current version.

>> Change has been made.

---

## Referee Report (RR1)

**Review**

Title: Nontrivial Relations Among Particle Collision, Relative Motion and Clustering in Turbulent Clouds: Computational Observation and Theory

Authors: Ewe-Wei Saw and Xiaohui Meng

**Comments**

The authors have rectified the queries raised by me. The manuscript looks better and may be considered for publication.

---

## Author Response (AR2)

Response to Anonymous Referee #2

The revised manuscript is significantly improved by the following:
1) simulations with gravitational settling have been included, thereby showing a preliminary view of the problem in a closer context to cloud physics.
2) A more thorough discussion and interpretation of the theory is included, significantly improving the clarity of the discussion.
3) The main manuscript has been carefully proofread, thereby reducing the number of distracting typos.
I recommend acceptance after dealing with a few additional corrections and clarifications.

>> We thank the referee for these positive remarks.

Note: line numbers refer to the "track-changes" version of the paper submitted with the revision.

Title: It started as "Nontrivial relations" and now it is "Intricate relations"… Not sure I'm a fan of either one because they are so vague. But they're not wrong, so it's the authors' call.

>> After much deliberations, we decide to keep the title unchanged.

Abstract: The abstract is thoroughly rewritten and is technically acceptable, but it must have missed the grammar check. Please correct throughout.

>> We have corrected the grammatical errors found.

Line 40: "dual" is confusing… do you mean "pair" or "two"?

>> we have revised it to "two".

Page 2, footnote 1: I am confused by the statement "without that condition, MRV is always zero in turbulence." A similar statement was made in the response to my review. My understanding is that inertial particles in the St<1 regime have a mean-inward drift velocity. Perhaps I am missing something here. The point is probably sufficiently subtle that it should be clarified in the main text rather than briefly mentioned in a footnote.

>> We thank the referee for raising this point. While revising this, we found that we had mistakenly considered (Sundaram& Collins, 1997) as a study of (non-colliding) ghost particles. In fact, it was a study on particle subjected to hard-sphere collisions. Thus, there should be no conditioning in the MRV they used. Also, the footnote is now unnecessary and will be deleted.

>> Besides this, the referee may have confused particles' mean relative velocity (MRV) with the so called "mean-inward drift velocity" which is related to the mean drift-flux (Eqn. 5 in the manuscript). We note that the latter results from correlation of turbulent fluctuations and should be interpreted as an additional mean inward velocity. In a system of ghost particles, MRV is zero, but under the separation paradigm, MRV conditioned on negative

velocities could be used to estimate collision rate. We have added a brief discussion of this at the end of Section 3.

Line 142: Correct "we shall we".
>> Corrected.

Lines 151-152: Why is this statement in quotation marks? Is it from another source? If yes, please clarify the source. If not, I'd recommend dropping the quotes to minimize confusion.
>> Corrected.

Line 155: I don't know what it means that the mean relative velocity "must be predominantly negative". It's either negative or not. Or do you mean over most of the range of scales r, or most of the Stokes number range, or something else? Also, how does this relate to the point made above regarding footnote 1 on page 2?
>> We have removed "predominantly" to avoid confusion. We have also removed "footnote 1 on page 2".

Page 13 and Figure 3: This is the most significant improvement to the paper and I applaud the authors for putting in the work to make these additional simulations to explore the effect of gravitational settling. As I expected, for the range of settling numbers up to order unity, there is no fundamental change in the behavior of the system. This greatly increases the relevance of the work to the cloud physics community. It might be worth noting that the range of Stokes numbers and settling parameters explored are very well aligned with measured values in natural clouds (e.g., Siebert et al. "Towards understanding the role of turbulence on droplets in clouds: In situ and laboratory measurements" Atmospheric Research, 2010).
>> The suggested reference has been added.

Line 297: Here "significant" might be an overstatement. I would say "from negligible to order 10%" or "from negligible to significant, but not dominant".
>> This change is now adopted.

Lines 304-308: The added discussion of hydrodynamic interactions and the reference to the recent work of Yavuz et al. 2018 is appreciated. Note that another paper just came out dealing with the same topic, so it may be worth citing: Bragg et al. "Hydrodynamic interactions and extreme particle clustering in turbulence" Journal of Fluid Mechanics, 2022.
>> The suggested reference has been added.

Line 317: Correct "This is could be".
>> Corrected.